# Metabolic Abnormalities Linked to Auditory Pathways in *ApoE*-Knockout HEI-OC1 Cells: A Transcription-Metabolism Co-Analysis

**DOI:** 10.3390/biom12091217

**Published:** 2022-09-01

**Authors:** Lu Ma, Hongshun Wang, Jun Yao, Qinjun Wei, Xin Cao

**Affiliations:** 1Department of Medical Genetics, School of Basic Medical Science, Nanjing Medical University, Nanjing 211166, China; 2Jiangsu Key Laboratory of Xenotransplantation, Nanjing Medical University, Nanjing 211166, China

**Keywords:** *ApoE*, HEI-OC1, glutamate metabolism, transcriptomics, metabolomics

## Abstract

Lipid transporter protein apolipoprotein E (APOE) has contributed to functional studies of various organ functions. Animals with *ApoE* knockout (KO) have been used to study atherosclerosis and hyperlipidemia while an increasing number of researchers have recently focused on the association of *ApoE* with hearing loss. A study found that *ApoE* KO mice experience sensorineural hearing loss and hair cell loss, but the exact mechanism is unclear. To explore the potential relationship between *ApoE* and hearing loss, we used HEI-OC1 cells (House Ear Institute-Organ of Corti) with Corti apparatus properties to reveal cell changes after *ApoE* knockout by combined transcriptome and metabolomic analysis. We found that glutamate deficiency, caused by reduced expression of glutamine transporter proteins, was a key correlate of basal metabolism and that inadequate glutamate causes apoptosis by reducing the cells’ resistance to external damage. Our study provides a reference mechanism for hearing loss due to *ApoE* KO.

## 1. Introduction

Apolipoprotein E (*APOE*) is secreted primarily by the liver and brain and is involved in lipid transport during enzymatic hydrolysis of fat and absorption into the blood. It interacts with members of the low-density lipoprotein receptor family to mediate the endocytic clearance of lipid proteins [1]. The effects of *ApoE* on tissues and organs are wide-ranging, with the most studied being the association of circulating *ApoE* with hyperlipidemia and atherosclerosis, and the in situ expressed *ApoE* with neurological pathologies such as Alzheimer’s disease and diseases such as hepatitis C and type 2 diabetes. In recent years, several population-based cohort studies have investigated the susceptibility to hearing loss associated with *ApoE* alleles, suggesting that age-related hearing loss (ARHL) is associated with *ApoE ε4* and the patients carrying this allele are predicted to have a two-fold increased risk of hearing impairment. The latest cross-sectional population studies and genome-wide association studies (GWASs) have demonstrated the implication of *ApoE* in ARHL [2,3]. In *ApoE* KO mice, researchers found that the animals had high-frequency hearing loss with 3–57% outer hair cell (OHC) and inner hair cell (IHC) deficits [4]; however, the mechanism by which *ApoE* affects hearing has not been reported.

HEI-OC1 is a conditionally immortalized cell line derived from H-2Kb-tsA58 transgenic mice that can represent Corti and express specific markers of cochlear hair cells and supporting cells [5,6]. As an ideal in vitro cell model for hair cell monomer research, HEI-OC1 is often used to study the cellular and molecular mechanisms involved in ototoxic-drug-induced deafness disease [7]. This tissue-specific cell line is particularly responsive to gene-protein expression profiles and metabolite composition changes.

Metabolomics confers the unique advantage of studying the time-dependent responses of genetically different individuals to environmental stimuli, and its identification of a wide range of small molecules in cells can be used to reconstruct the current state of the true cellular response, which is widely employed in the study of molecular mechanisms of disease and biomarker discovery. Recently, metabolomics has been used to study the mechanisms involved in the genetic dependence of different *ApoE* subtypes in Alzheimer’s disease, but there is still a lack of adequate research data on *ApoE*-deficient disease models. The transcriptome is well established for determining gene expression profiles in disease states and can identify potential changes in the genetic code. With the growth of combined multi-omics analysis, the integration of the transcriptome and metabolome may further improve our understanding of the regulation of gene expression with respect to cellular phenotypes.

In this study, we combined liquid-chromatography-mass-spectrometry (LC-MS) non-targeted metabolomics and transcriptomics methods to investigate the altered activity of HEI-OC1 cells after *ApoE* KO.

## 2. Materials and Methods

### 2.1. Cell Culture

*ApoE*-KO and wild-type (WT) HEI-OC1 cells (House Ear Institute-Organ of Corti cells, Los Angeles, CA, USA) were cultured in Dulbecco’s modified Eagle’s medium (DMEM, Gibco, Grand Island, NE, USA) with 10% fetal bovine serum (#FSP500, ExCell Bio, Shanghai, China) in a 33 °C cell incubator with 10% CO_2_. Cells were digested with trypsin after 80% growth of 10 cm dishes, and 1 × 10^6^ cells were counted and collected into centrifuge tubes. After washing with phosphate-buffered saline, the supernatant was discarded, shock frozen in liquid nitrogen, and stored at −80 °C.

### 2.2. Generation of ApoE-Deficient HEI-OC1 Cell Lines

Single RNAs (sgRNAs) were designed for the third exon of the mouse *ApoE* gene and were ligated into the PX330 plasmid containing the Cas9 (CRISPR, clustered regularly interspaced short palindromic repeats, associated protein 9) backbone. The recombinant plasmids were transfected into HEI-OC1 cells using the Amaxa™ Basic Nucleofector™ kit (Lonza, Cologne, Germany) and viable monoclonal cell colonies were obtained by screening for G418 (geneticin, aminoglycoside antibiotic) (#A1720, Sigma, Taufkirchen, Germany). Genotyping was performed by polymerase chain reaction (PCR) sequencing (Appendix A). ApoE knockout efficiency was verified with Western blotting (WB).

### 2.3. Intracellular ROS Assay

To detect intracellular reactive oxygen species levels, using a ROS detection kit (KeyGEN Biotech, Nanjing, China), dichlorfluorescein-discrimination analysis (DCFH-DA) was diluted with serum-free culture medium to reach a final concentration of 10 μM, added into 6-well plates containing approximately 5 × 10^6^ cells/well, and placed in a 33 °C cell incubator for 20 min. After 3 washes with PBS, the cells were harvested and placed in a flow cytometer to detect fluorescence intensity. Positive controls included ROS stimulating agent (Rosup) (50 μg/mL, 30 min).

### 2.4. Apoptosis Detection

Adherent cells planted in the same number were washed twice after treatment according to the conditions. Single free cells were obtained after digestion with trypsin without ethylenediaminetetraacetic acid for 2 min. Cells were stained according to the steps of the Annexin V-FITC/PI Apoptosis Detection Kit (Vazyme, Nanjing, China). In the flow analysis, experimental groups for Annexin V-Fluorescein isothiocyanate (FITC) and propidium iodide (PI) mono coloring were necessary to perform regulatory compensation.

### 2.5. Total RNA Extraction and Quantitative Real-Time Reverse Transcription PCR (qRT-PCR)

Cells were homogenized in guanidine thiocyanate (Life Technologies, Carlsbad, CA, USA) using the phenol-chloroform extraction method to obtain total RNA and complementary DNA (cDNA) was synthesized via a TransScript^®^ One-Step gDNA Removal and cDNA Synthesis SuperMix kit (TransGene Biotech, Beijing, China). Real-time PCR was performed using the PerfectStart^®^ Green qPCR SuperMix (+Dye I) kit (TransGene Biotech, Beijing, China) on a Step One Plus Real-Time PCR System (Applied Biosystems, Waltham, MA, USA). The experiment was repeated with three biological samples to obtain the mean gene expression levels, and the comparative cycle of quantification (CT) method (2^−ΔΔ^CT) was used to calculate the relative gene expression. Glyceraldehyde 3-phosphate dehydrogenase (GAPDH) served as an internal control. The primer sequences are listed in the Appendix A.

### 2.6. Transcriptome Analysis

RNA quantification and qualification were performed using the RNA Nano 6000 Assay Kit of the Bioanalyzer 2100 system (Agilent Technologies, Santa Clara, CA, USA), and library preparation for transcriptome sequencing using the AMPure XP system (Beckman Coulter, Pasadena, CA, USA) was used to obtain purified PCR products, which were quantified by a Qubit 2.0 Fluorometer and qRT-PCR. Different libraries were sequenced by the Illumina NovaSeq 6000 (Illumina, San Diego, USA), and clean data (clean reads) were obtained by removing low-quality reads from raw data and calculating the quality score (Q20, Q30) and guanine-cytosine content (GC content). Differential expression analysis of the two groups was performed using the DESeq2 R package (1.20.0) (differential expression analysis for sequence data version 2R, Bioconductor). The resulting *p* values were adjusted using Benjamini and Hochberg’s approach for controlling the false discovery rate. *p* ≤ 0.05 and |log2 (foldchange)| ≥ 1 were set as the thresholds for significantly different expression. 

### 2.7. Metabolomics Analysis 

The metabolite extraction solution was prepared using methanol and acetonitrile, which were added to an isotopically labeled internal standard mixture. The samples were frozen and thawed 3 times in liquid nitrogen and sonicated in an ice-water bath. The samples were incubated at −40 °C and analyzed with the centrifuged supernatant. The quality control (QC) sample was prepared by mixing an equal aliquot of the upper layer from each specimen. LC-MS/MS chromatographic separation was implemented on an ultra-high-performance liquid chromatography (UHPLC) system (Vanquish, Thermo Fisher Scientific, Waltham, MA, USA) with a UPLC Ethylene Bridged Hybrid Amide column (2.1 × 100 mm, 1.7 μm) coupled to an Orbitrap Exploris 120 mass spectrometer (Orbitrap MS, Thermo). To reduce the influence of detection system error on the results, raw data were managed based on the relative standard deviation (RSD) to filter irrelevant values, and normalized data were acquired based on an internal standard (IS). A total of 9709 peaks were retained after preprocessing and served as a basis for subsequent research. Variable important in projection (VIP) >1, *p* value ≤ 0.05, and |log2 (foldchange)| ≥ 1 were set as the thresholds for significantly different expression. 

## 3. Results

### 3.1. Quality Analysis of Metabolome and Transcriptome Data

After quality control and pre-processing of the data, the data characteristics of the two sample groups were assessed. Principal component analysis (PCA) of the metabolomic data from the six groups of samples sent for testing, after dimensionality reduction treatment, showed that WT and KO cells clustered within the same group of samples and were significantly different between samples of different groups, indicating good reproducibility of the samples for testing, which is the basis for representative results (Appendix A). After grouping the data from both clusters, a supervised analysis of orthogonal partial least squares discrimination analysis (OPLS-DA) was performed, and component 1 accounted for 58.5%, indicating that the composition of the KO-treated cell data was significantly different from that of WT, which was the foundation for further analysis of variance (Figure 1A). To monitor the overall performance of the individual sample data, correlation coefficients between samples were calculated using Pearson’s method and data from samples with large outliers were excluded (Figure 1B). The experimental data from all six samples at this time were included in the subsequent analysis.

The resulting three transcriptome data pairs were analyzed as described above. PCA analysis showed standardized treated data scattered on the plane according to their characteristics, and differences between treated samples were determined according to the PC1 content (Figure 1C). The inter-sample correlation coefficient scores are shown in each box (Figure 1D).

### 3.2. Changes in Cell Metabolism after ApoE KO

The metabolites identified were classified according to the human metabolic database (HMDB, University of Alberta, Edmonton, AB, Canada). There are 13 items annotated under the superclass while many belong to ‘lipids and lipid-like molecules’, accounting for 37%; followed by ‘organic acids and derivatives’, accounting for 24.5%; ‘organoheterocyclic compounds’, accounting for 11.9%; and several other categories, accounting for less than 10%. There are obvious dominant components under the subclass of the first two categories, including 133 amino acids peptides and analogues, 87 glycerophosphocholines, 40 fatty acids and conjugates, and 25 glycerophosphoethanolamines (Figure 2A).

Further study of the 133 metabolites altered in amino acids peptides and analogues revealed that the aminoacyl-tRNA biosynthesis pathway was significantly downregulated due to the absence of raw materials (Figure 2B). Aminoacyl-tRNA is a tRNA bound to the corresponding amino acid that delivers amino acids to the ribosome and adds them to extend the polypeptide chain. The lack of amino acids such as L-phenylalanine, L-arginine, L-glutamine, glycine, L-serine, L-methionine, L-valine, L-alanine, L-threonine, tyrosine, and L-glutamic acid suggests that intracellular protein synthesis is blocked or in the early stage of becoming blocked (Figure 2C). 

The metabolites under the ‘lipids and lipid-like molecules’ classification are mainly concentrated among glycerophosphocholines, including the rearrangement of the internal composition ratios of phosphatidyl ethanolamine (PE), phosphatidylcholine (PC), and Lyso (Figure 2D). The content of most substances in the same classification increased. PC and PE are involved in the cell membrane and sub-organelle membranes, such as those of the mitochondria and Golgi. PC is the most abundant phospholipid in the cell membrane, PE has the highest proportion in the mitochondria, and the ratio of PC to PE is related to the liquidity of the phosphatidic lipid bilayer. At the same time, changes in the mitochondrial membrane component can affect the organelle electron transport chain and ATP production, and in endoplasmic reticulum membranes, an alternation of glycerophospholipid components may be observed under oxidative stress. Changes in LysoPC and LysoPE ratios were observed simultaneously.

### 3.3. L-Glutamic Acid Is the Central Substance of Metabolic Changes in KO Cells

Detected metabolites and genes were further tested against the thresholds, and substances with a difference of more than two-fold were included in further analysis. In total, 3595 upregulated and 6186 downregulated differential metabolites were identified among the detected metabolites. In the transcriptome, 639 upregulated and 1079 downregulated differentially expressed genes were obtained (Appendix A).

The altered metabolic pathways reflect the effects of gene knockout on cells. Functional enrichment of each annotated differential metabolite was performed, and histograms of the enrichment ratios showed both high enrichment indices and highly statistically significant differences for the D-glutamine and D-glutamate metabolism pathways (Figure 3A). This means that multiple metabolites are affected and thus altered in this pathway, suggesting it might be the most affected metabolic pathway in KO cells. In total, five substances were detected: D-glutamine, L-glutamine, DL-glutamate, L-glutamic acid, and oxoglutarate were all significantly reduced in KO cells (Appendix A). A network of metabolite relationships was constructed on the Metamapp platform (University of California, Davis, CA, USA), and a map of the correlations between the five metabolites was visualized using Cytoscape (Cytoscape Consortium, University of California, San Francisco, CA, USA, University of California, San Diego, CA, USA, University of Toronto, Toronto, ON, Canada, Gladstone Institutes, San Francisco, CA, USA, Agilent Technologies, Santa Clara, CA, USA, Institute Pasteur, Paris, France, Institute for Systems Biology, Seattle CA, USA) (Figure 3B and Appendix A). L-glutamic acid is a common product of the other four metabolites, implying that it may be a central link in the D-glutamine and D-glutamate metabolism pathways. 

The network of all the different metabolites detected was plotted against L-glutamic acid, allowing the integration of scattered metabolite data while finding upstream and downstream metabolite relationships. All differential metabolites related to glutamate, including intermediates of glutamate metabolism, several amino acids involved in life processes, and critical metabolites participating in the tricarboxylic acid (TCA) cycle, urea cycle, and redox homeostasis, were downregulated in KO cells (Figure 3C). Key enzymes encoded by genes are also involved in metabolite changes. Metascape was used to map the overall network of differential gene–metabolite associations. A reciprocal network related to L-glutamic acid metabolism was found in the urea cycle and metabolism of the arginine, proline, glutamate, aspartate, and asparagine pathway (Figure 3D). The relationship map equally indicated that many more genes regulate L-glutamic acid than L-glutamine and oxoglutarate, supporting the idea that L-glutamic acid is a key substance in altered cellular metabolism, but that it is a metabolic process regulated by multiple genes.

Based on metabolite networks and gene association networks, altered pathways of L-glutamic acid metabolism were mapped in KO cells (Figure 3E). Intermediate metabolites upstream of the conversion of proline, histidine, and arginine to L-glutamic acid were reduced, and the expression of the *Aldh4a1* (aldehyde dehydrogenase family 4, member A1) gene, which encodes an enzyme in this process, was also reduced. Most importantly, the expression of key enzymes related to L-glutamic acid cross-conversion was upregulated; in contrast to the three metabolites that were reduced together, the expression of most of these genes was elevated, including *Gls* (glutaminase), the main enzyme gene encoding the breakdown of glutamine into glutamate, and *Got* (glutamic oxaloacetic transaminase) and *Gpt2* (glutamine-pyruvate transaminase 2), the main transaminase enzyme genes (Appendix A). Interestingly, the enzymes encoded by these genes are mainly expressed in mitochondria, probably due to a compensatory increase caused by insufficient glutamate in mitochondria.

### 3.4. Decreased Expression of Slc7a8 and Slc6a19 Is Responsible for Abnormal L-Glutamic Acid Metabolism in KO Cells

The significantly altered glutamate metabolic pathway revealed through associated gene interaction analysis suggested a possible compensatory response in the cell, but the direct cause of the overall decline in the whole metabolic pathway remains unclear, so we further investigated the cellular transcriptomic data. By analyzing the KEGG (Kyoto Encyclopedia of Genes and Genomes) pathways of downregulated differentially expressed genes, we identified the protein digestion and absorption pathway with a high enrichment index (Figure 4A) while GSEA (Gene Set Enrichment Analysis) also suggested a significant downregulation of this pathway (Appendix A). All genes in this pathway were analyzed in relation to five metabolites within the metabolic pathway (Figure 4B). By excluding most of the genes encoding collagen and the proton pump, we identified *Slc7a8* (solute carrier membrane transport protein, family 7, subfamily A, member 8, large neutral amino acids transporter) and *Slc6a19* (solute carrier membrane transport protein, family 6, subfamily A, member 19, sodium-dependent neutral amino acids transporter), which were validated by qPCR and were associated with metabolites (Appendix A).

*Slc7a8* and *Slc6a19* are neutral amino acid transport proteins located in the cell membrane and are responsible for transporting substances such as glutamine from the extracellular to the intracellular compartment. Glutamine is considered to be the fuel of the cell; in addition to catabolism, as glutamate participates in amino acid metabolism, the urea cycle, and the TCA cycle, it is also a source of free amines and serves as the raw material for glutathione in the cytoplasm. Reduced *Slc7a8* and *Slc6a19* expression indicates that glutamine entry into the cell is reduced, which affects cell metabolism as glutamine is one of the most important sources of glutamate (Figure 4C).

### 3.5. Increased Intracellular Oxidative Stress in KO Cells

ROS are a by-product of the cellular oxidative respiratory chain, and under normal physiological conditions, when ROS clearance and production are in balance, cells can respond to temporary external stimuli and perform a protective function. Glutathione is an important substance responsible for intracellular ROS clearance and is composed of glutamate, cysteine, and glycine. KO of intracellular glutamate and glycine deficiency can lead to ROS accumulation while many studies have also shown a correlation between *ApoE* KO and increased ROS.

We next used flow cytometry and immunofluorescence to detect the intracellular ROS content. As expected, *ApoE* KO indeed increased the intracellular ROS content compared with normal levels (Figure 5A,B).

Because of the reduced metabolic activity in KO cells, we speculated whether *ApoE* KO would produce a weaker response to increased external stimuli. We exposed the cells to 200 or 400 μM H_2_O_2_ for 3 h to construct a transient oxidative damage environment and calculated the number of positive cells under a flow cytometer (Figure 5C). The ROS peaks of WT and KO cells showed a more obvious right shift in 200 μM H_2_O_2_; however, at 400 μM H_2_O_2_, the KO peak was no longer shifted to the right, indicating that intracellular ROS were overloaded.

### 3.6. Elevated Apoptotic Index after KO Cell Stress

Excessive levels of ROS can cause damage to proteins, nucleic acids, membranes, organelles, etc., and thus activate the process of apoptosis. When the cells were cultured, a large number of dead cells appeared together with proliferative activity. The cells were cultured in the same numbers and incubated continuously without changing the medium or washing over a period of two days, during which time the cells lost their morphology and the number of dead cells gradually increased (Figure 6A).

To understand the ability of cells to resist damage leading to death under stress, 0, 200, and 400 μM H_2_O_2_-gradient treatment controls were also established, and the numbers of cells with early and late apoptosis were determined by flow cytometry, with cells showing more signs of late apoptosis after stress (Q2) compared to early apoptotic cells, which exhibited no significant fluctuations (Q3) (Figure 6B). Therefore, a line graph was plotted for the number of late apoptotic cells, and the baseline apoptosis value for cells after knockdown was higher than that for WT cells. With increasing H+ stimulation, the number of late apoptotic cells increased in both groups. Based on the slope of the line, it was evident that the apoptosis index of co-cells increased after stress.

## 4. Discussion

Previous studies on the function of the *ApoE* gene have focused on its implication in atherosclerosis and hyperlipidemia. However, the implication of *ApoE* in hearing loss has not been thoroughly elucidated. The cochlea is one of the few terminal organs lacking blood vessels, in particular the hearing-generating Corti apparatus, which has no direct source of blood supply and is almost unexposed to lipids; thus, we used the HEI-OC1 cell line with cochlear hair cell properties under culture conditions without supplemental lipids to predict the role of the *ApoE* gene in specific tissues, such as the Corti apparatus, and describe possible mechanisms leading to hair cell loss after *ApoE* KO.

In our study, we identified a significantly altered intracellular glutamate and glutamine pathway while an altered glutamate content was also detected in the cochlear-ablation rats [8]. L-glutamic acid is a by-product of other metabolites in this pathway and a vital intermediate that influences metabolic activity through deamidation, decarboxylation, and involvement in enzyme and hormone synthesis [9]. Of the differential metabolites we studied that interact with glutamate, deficiency of several non-essential amino acids leaves insufficient raw material for the synthesis of aminoacyl-tRNA while the reduction in essential metabolites of the tricarboxylic acid and urea cycle intermediates suggests that low levels of metabolic activity persist in glutamate-deficient *ApoE* KO cells. Furthermore, the glutamate decarboxylation product, the brain inhibitory neurotransmitter gamma aminobutyric acid (GABA), is also present at low levels in KO cells, which overlaps with population census results and may provide new insights into the role of *ApoE* genes in neurological disorders [10]. Among the differential metabolites we identified, the presence of oxidized glutathione, a side-product of the intracellular glutathione redox reaction, was significantly reduced, and the same reduction was observed in the elderly population, which may be predictive of premature aging [11]. Thus, glutamate has wide-ranging effects on cellular activity, mainly affecting intracellular basal metabolism.

Glutamine is a precursor of glutamate, which plays an important role in transporting nitrogen between tissues and is not an essential amino acid but has been redefined as a conditionally essential amino acid to provide nutrition to reduce cell death when the body is under stress [12]. Thus, the increased need for glutamine is met by transporter proteins in the cell membrane [13]. The sodium-dependent neutral amino acid transporter encoded by *Slc6a19* causes reduced glutamine uptake after inhibition [14,15] while the L-type amino acid transporter protein 2 (LAT2) encoded by the *Slc7a8* gene is responsible for the transport of neutral amino acids, including glutamine: the levels of both of these proteins are reduced in KO cells. It has been shown that the overexpressed *Slc7a8* gene can regulate glutamine-dependent mTOR (mammalian target of rapamycin, member of the phosphatidylinositol 3-kinase-related kinase family of protein kinases) activation in cancer, which also requires high glutamine consumption, suggesting its role in glutamine transport [16]. *Slc7a8* has also been demonstrated to be a novel gene involved in age-related hearing loss, as suppression of this gene causes damage to the Corti apparatus, spiral ganglia, and vascular striae [17]. Direct interaction of *ApoE* with *Slc7a8* or *Slc6a19* has not been shown, but the published literature clearly shows that *ApoE* binds to several membrane surface receptors, such as LDLR: thus, we hypothesize that *ApoE* may be involved in intracellular glutamate metabolism by binding to membrane receptors and influencing signal transduction to promote *Slc7a8* and *Slc6a19* expression.

The accumulation of ROS in KO cells further confirms that the cells were under stress, which is consistent with most published studies and reflects an increased cellular demand for glutamine, but the difference in the number of apoptotic cells under natural conditions was not significant, and increased environmental stimuli corresponded to an increased late apoptosis index in KO cells. In conclusion, we hypothesize that *ApoE* KO indirectly induces programmed cell death by affecting glutamate metabolism in HEI-OC1 cells and weakening their resistance to external damage. Based on this study, moderate supplementation of glutamine may have a protective effect in the treatment or prevention of hearing loss, especially in elderly patients with age-related accumulation of reactive oxygen species. 

In this research, we combined advanced metabolomics and transcriptomic techniques to investigate the role of the *ApoE* gene in auditory HEI-OC1 cells at the molecular level. We found that *ApoE* regulates the intracellular glutamate content and REDOX balance by affecting the expression of cell membrane transporter genes *Slc7a8* and *Slc6a19*, leading to *ApoE* KO hair cells undergoing apoptosis after mild stimulation, which may be the mechanism of hearing loss in KO mice. It also provides a reference molecular basis for studying the relationship between *ApoE* dysfunction and deafness in the human population. However, our results need to be confirmed by in vivo experiments, which is a major limitation of this study.

## Figures and Tables

**Figure 1 biomolecules-12-01217-f001:**
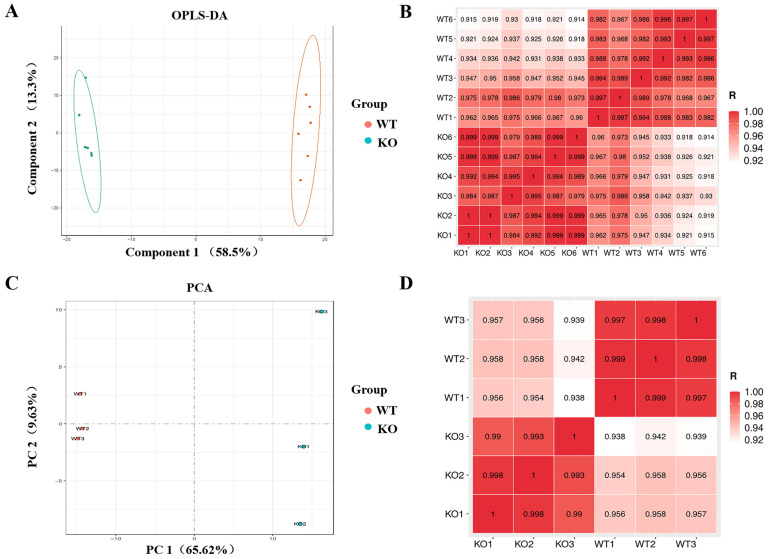
Assessment of data quality for metabonomic and transcriptome analysis. (**A**) Metabolite partial least squares discriminant analysis (OPLS-DA) (*n* = 6). (**B**,**D**) Correlation analysis between samples, with colors indicating the strength of correlation. (**C**) Transcriptomics data principal component analysis (PCA) (*n* = 3).

**Figure 2 biomolecules-12-01217-f002:**
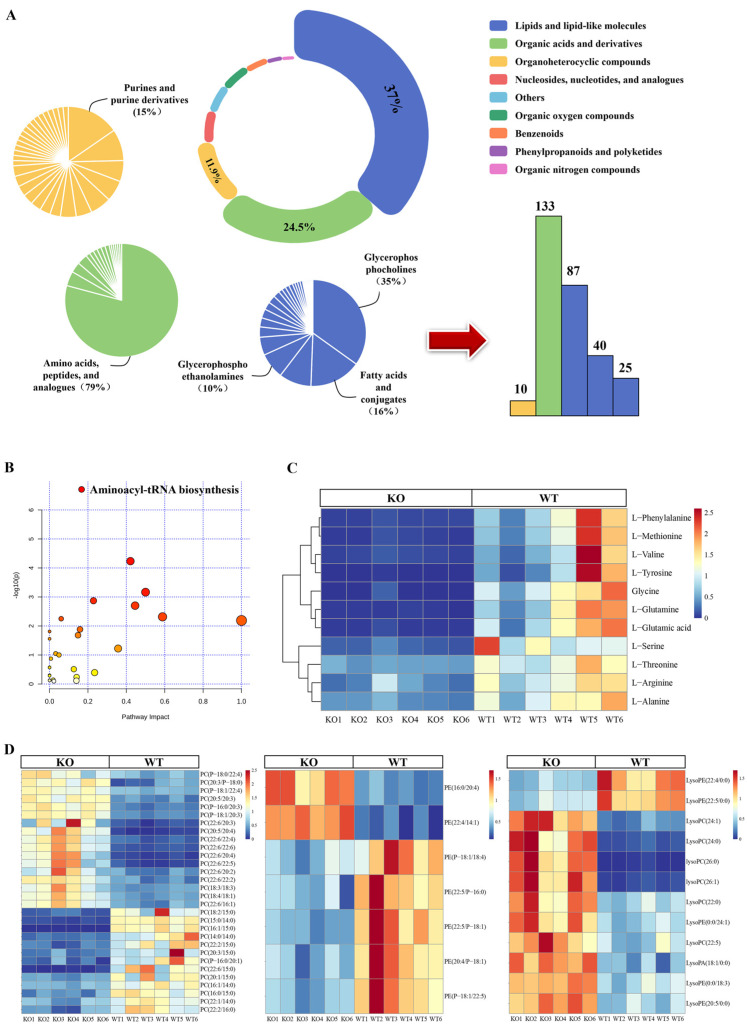
Identification of differential metabolites in KO cells. (**A**) Structural composition of differential metabolites. The composition and quantity distribution of the top three metabolites are displayed. (**B**,**C**) Functional annotation of amino acids, peptides, and analogues metabolites. Metabolite enrichment showed that aminoacyl-tRNA was the most severely affected, and all the detected metabolites were downregulated in the aminoacyl-tRNA pathway. (**D**) The altered glycerophosphocholine is mainly PE, LysoPE, LysoPC, and PC. Metabolites in this classification were both up- and downregulated in the cell, with upregulation predominating. On this heatmap, each box represents one substance, with red and blue representing up- and downregulation, respectively, and the darker the color, the greater the difference.

**Figure 3 biomolecules-12-01217-f003:**
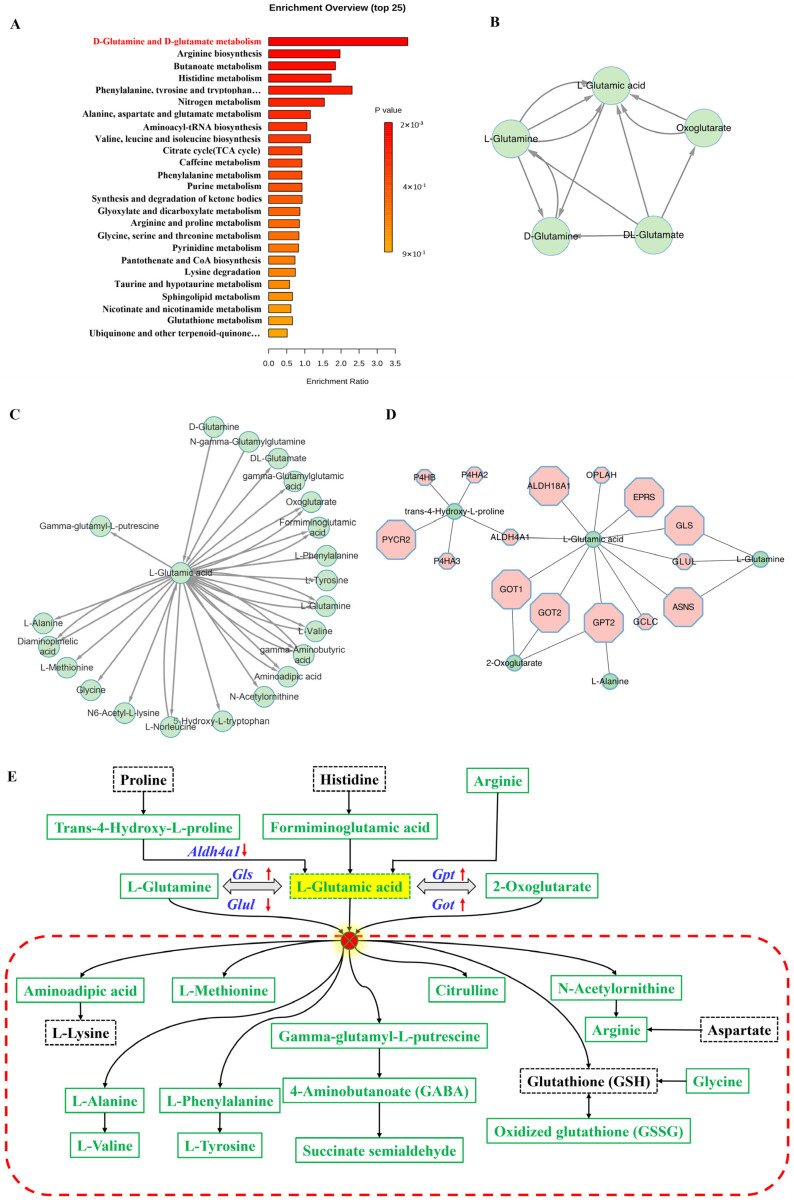
D-glutamine and the D-glutamate metabolism pathways and associated gene expression were affected. (**A**) Analysis of all metabolites with significant differences. The metabolism enrichment index of D-glutamine and D-glutamate metabolism was highest in metabolic activities (*p* < 0.005). The higher the enrichment index, the more obviously the pathway was affected, and *p* < 0.05 indicated that the result was statistically significant. (**B**) Five substances in the pathway were downregulated. In the metabolite association analysis, all four metabolites were associated with the production of L-glutamic acid. (**C**,**D**) Map of all L-glutamic-acid-related differentiated metabolites and gene networks in metabolite association analysis (**C**) and metabolite (green circle)–gene (red hexagon) association analysis (**D**). (**E**) Pattern diagram of integrated metabolite–gene interaction network. Green indicates metabolites that are detected and also reduced, and black indicates metabolites with insignificant changes. Blue represents genes, with arrows pointing to suggest trends. Other colours without special instructions do not imply trend change. (*Aldh4a1*, aldehyde dehydrogenase family 4, member A1, *Gls*, glutaminase, *Got*, glutamic oxaloacetic transaminase, *Gpt*, glutamine-pyruvate transaminase, *Glul*, glutamate-ammonia ligase).

**Figure 4 biomolecules-12-01217-f004:**
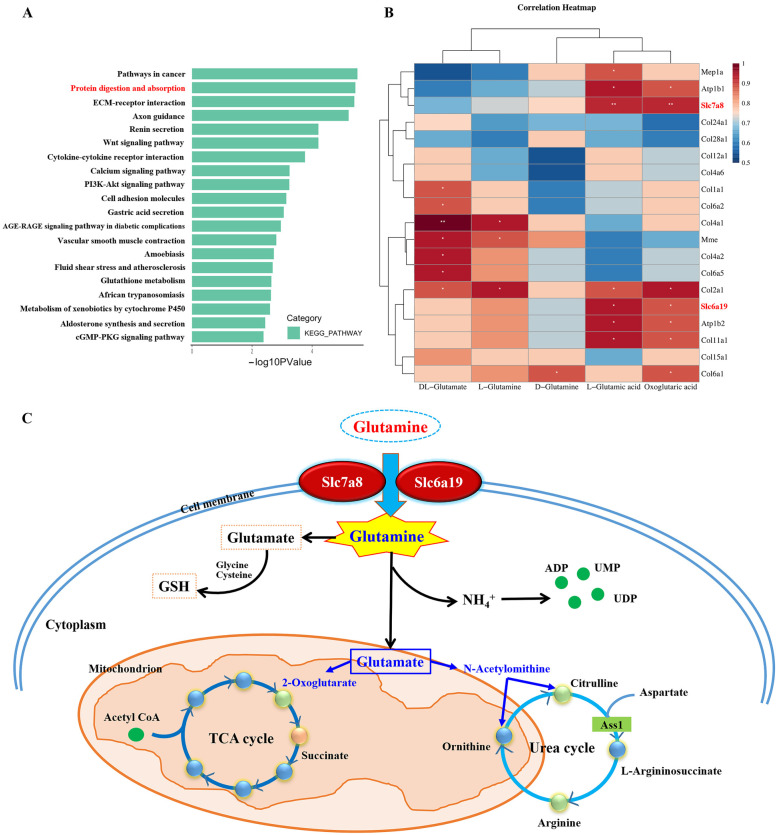
*Slc7a8* and *Slc6a19* influence the D-glutamine and D-glutamate metabolism pathways by reducing the influx of glutamine into cells (*Slc7a8*, solute carrier membrane transport protein, family 7, subfamily A, member 8, large neutral amino acids transporter, *Slc6a19*, solute carrier membrane transport protein, family 6, subfamily A, member 19, sodium-dependent neutral amino acids transporter). (**A**) KEGG analysis of significantly downregulated genes showed protein digestion and absorption pathway enrichment. (**B**) Association analysis of genes in the protein digestion and absorption pathway with products involved in D-glutamine and D-glutamate metabolism. *Slc7a8* and *Slc6a19* encode proteins that mediate transmembrane import of glutamine that are significantly associated with metabolites. (**C**) Scheme of metabolism in KO cells. Spearman’s test was used for correlation calculation, ** *p* < 0.01, * *p* < 0.05 (PI3K, phosphorinositide 3-kinaseS, cGMP, cyclic guanosine monophosphate, PKG, protein kinase G, GSH, glutathione, ADP, adenosine diphosphate, UMP, uridine monophosphate, UDP, uridine diphosphate, and mention the meanings of col, collagen, *mep*, 2-C-Methyl-D-erythritol 4-phosphate, *mme*, membrane metalloendopeptidase, ATP, adenosine triphosphate).

**Figure 5 biomolecules-12-01217-f005:**
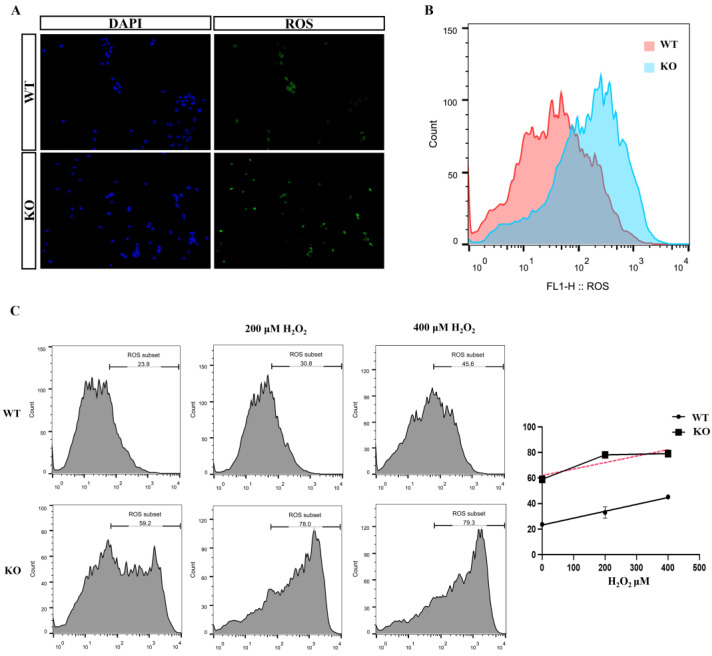
Elevated ROS content and decreased reserve capacity in KO cells. (**A**,**B**) Increased numbers of ROS-positive cells (green) in KO cells and rightward shift of ROS peak by flow cytometry. (**C**) The right shift of the intracellular ROS peak in 0, 200, 400 and μM H_2_O_2_ state is shown in the graph. The red dotted line represents the slope. (DAPI, 4′, 6-Diaminidin-2-phenylindol, ROS, reactive oxygen species).

**Figure 6 biomolecules-12-01217-f006:**
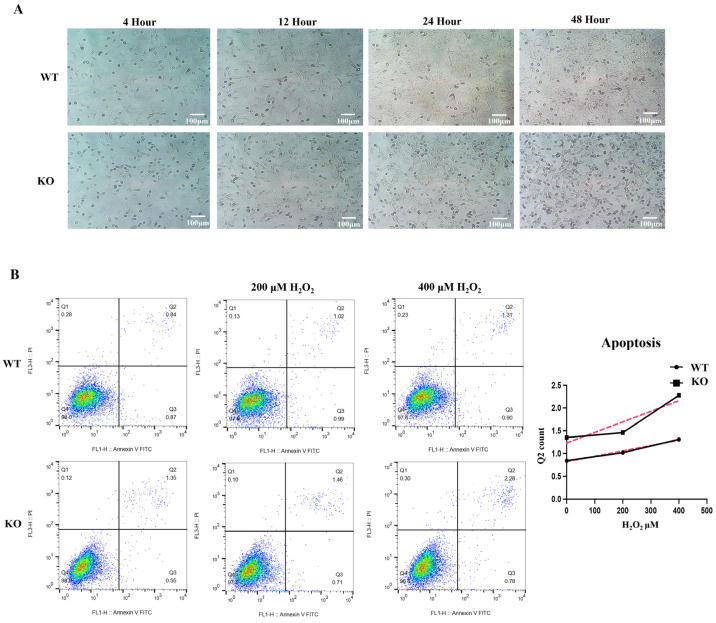
Raised index of late apoptosis in KO cells. (**A**) KO cell proliferation is accompanied by increased production of dead cells. In total, a 5 × 10^5^ amount of cells were cultivated in 33 °C 10% CO_2_ in 6-well plates and observed and photographed every 12 h under a phase-contrast microscope for 2 days. (**B**) Apoptosis was detected by flow cytometry. The number of positive cells in the Q2 quadrant, representing the late stage of apoptosis, was counted to plot the line graph. Cells were cultured under 0, 200, and 400 μM H_2_O_2_ for 3 h before the test. The red dotted line represents the slope.

## Data Availability

Not applicable.

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
