# Peer review of "Metabolic Abnormalities Linked to Auditory Pathways in ApoE-Knockout HEI-OC1 Cells: A Transcription-Metabolism Co-Analysis"

_biomolecules, 2022, doi:10.3390/biom12091217_

Round 1
Reviewer 1 Report
In their manuscript "metabolic abnormalities of ApoE knockout in auditory of HEI-OC1 cells: a transcription-metabolism con analysis" Lu Ma et al. present their experimental data of metabolomics and transcriptomics in HEI-OC1 cells after ApoE knockout. As a main results they discuss the significantly altered intracellular glutamate and glutamine pathway as a possible reference mechanism for HL in ApoE knockout HEI-OC1 cells.
Overall the manuscript is well written, scientifically sound and adds to the literature. English is appropriate, the conclusions are supported by the data and the discussion focusses on the most important findings of the study. Except some minor points which should be adressed before acceptance the reviewer recommends the presented manuscript for publication in MDPI biomolecules:
Minor points:
a) Titel: the reviewer recommends to change the titel as it is somehow missledaing. E.g. "matabolic abnormalities linked to auditory pathways in ApoE KO of HEI-OC1 cells: atranscriptio-metabolism co-analysis" or "Altered intracellular glutamate and glutamin pathway in ApoE KO of HEI-OC1 cells: a reference mechanism of ApoE KO HL?" etc.
b) Introduction section: The introduction should comment on the type of HL associated with E4 and its evolution.
c) Page 2, Lines 45-47: The whole sentence is hard to read. Please rewrite and shorten.
d) Discussion: Is an altered intracellular glutamate and glutamine pathway described in other models of HL so far? This should be discussed.
e) Please comment on potential therapeutic and/or prophylactic implications of your findings for ApoE associated HL.
Author Response
Responding to Reviewer 1
Thank you for your constructive comments on our manuscript. Your comments are valuable for improving our present experiment. The manuscript has been revised accord to your comments. The details of revisions are listed as below.
----------------------------------------------------------------------------------------------------------------
a) Title: the reviewer recommends to change the title as it is somehow misleading. E.g. "metabolic abnormalities linked to auditory pathways in ApoE KO of HEI-OC1 cells: a transcriptiom-metabolism co-analysis" or "Altered intracellular glutamate and glutamine pathway in ApoE KO of HEI-OC1 cells: a reference mechanism of ApoE KO HL?" etc.
Answer: The title of the article has been changed.
b) Introduction section: The introduction should comment on the type of HL associated with E4 and its evolution.
Answer: We have added the association between E4 and the type of HL in Paragraph 1of Introduction.
c) Page 2, Lines 45-47: The whole sentence is hard to read. Please rewrite and shorten.
Answer: As you suggested, we decided to remove this ambiguous sentence.
d) Discussion: Is an altered intracellular glutamate and glutamine pathway described in other models of HL so far? This should be discussed.
Answer: Previous studies have also found reduced levels of glutamate in the brain of rats with HL after cochlear ablation, and we have added part of the content and corresponding citation in the second paragraph of the Discussion.
e) Please comment on potential therapeutic and/or prophylactic implications of your findings for ApoE associated HL.
Answer: We have added the comments in the end of Paragraph 4 of the Discussion on the possible therapeutic or preventive value of glutamine in hearing loss.
Reviewer 2 Report
Dear Ladies and Gentlemen, Dear Journal-Team,
the interesting manuscript 'Metabolic abnormalities of ApoE knockout in auditory HEI-OC1 cells: a transcriptom-metbolism co-analysis' investigates the metaobolic and transcriptomic interactions evoked by ApoE knockout. It is well planned. Tables and Figures are sufficient. After revision I am happy to review the manuscript again.
a) Please include a conclusion, wherein parts of the well discussed results from the discussion section and the last nine lines of the introduction can be shifted.
b) Make sure that the letter size in the figures and supplementary figures are high or big enough that it can be read without difficulties in any software format.
c) Explain in the legends the p-values in Figure 3 and the curve of the enrichment score of Figure S6. Is it correct that in Figure 1A component 1 is on both axes and in Figure 1C PC1 and PC2?
d) Mention of every software or materials used the producer with its main residence: HEI-OC1 cells (House Ear Instituts-Organ of Corti cells, Los Angeles, USA), Omicsstudio (introduction, line 12 from the bottom), Amaxa basic nucleofector kit (Lonza, Cologne, Germany, section, Generation of ApoE-deficient HEI-OC1 cell lines), KeyGen Biotech (Nanjing, China, section Intracellular ROS assay), Vazyme (Nanjing, China, section, Intracellular ROS assay), Qubit fluorometer (section Transcriptome analysis), Illumina (San Diego, USA, section Transcriptome analysis), DESeq2 R package (section Transcriptome analysis), UHPLC system (Vanquish, Thermo Fisher Scientific, Waltham Massachusetts, USA, section Metabolomics analysis), Orbitrap (section Metabolomics analysis), HMDB (human metabolome database, University of Alberta, Canada, section Changes in cell metabolism after ApoE KO), Metamapp platform (University of California, Davis, USA, section, L-Glutamic acid is the central substance of metabolic changes in KO cells), Cytoscape (Cytoscape Consortium, University of California, San Francisco, USA, University of California, San Diego, USA, University of Toronto, Canada, Gladstone Institutes, San Francisco, USA, Agilent Technologies, Santa Clara, USA, Institut Pasteur, Paris, France, Institute for Systems Biology, Seattle USA, section, L-Glutamic acid is the central substance of metabolic changes in KO cells).
e) The manuscript needs to be thoroughly revised for mentioning all abbreviations, when first used: HEI-OC1 (abstract), KO (knockout, abstract), OHC, IHC (outer hair cells, inner hair cells, line 13 of the introduction), ROS (reactive oxygen species, line 3 from the bottom of the introduction), genes Slc7a8, Slc6a19 (explain them already in the introduction and not in the discussion, Slc7a8=solute carrier membrane transport protein, family 6, subfamily a, member 8, large neutral amino acids transporter, Slc6a19=solute carrier membrane protein, family 6, subfamily a, member 19, sodium-dependent neutral amino acid transporter), LC-MS (liquid-tomography-mass-spectrometry, introduction, line 9 from the bottom), FBS (fetal bovine serum, section cell culture), PBS (phosphate buffered saline, section cell culture), Cas9 (CRISPR, clustered regularly interspaced short palindromic repeats, associated protein 9, section Generation of ApoE-deficient HEI-OC1 cell lines), G418 (geneticin, aminoglycoside antibiotic, section Generation of ApoE-deficient HEI-OC1 cell lines), WB (western blot, section Generation of ApoE-deficient HEI-OC1 cell lines), PCR (polymerase chain reaction, section Generation of ApoE-deficient HEI-OC1 cell lines), DCFH-DA (dichlorfluorescein-discrimination analysis), Rosup (ROS stimulating agent, section intracellular ROS assay), EDTA (Ethylenediaminetetraacetic acid, section apoptosis detection), FITC (Fluorescein isothiocyanate, section intracellular ROS assay), PI (section, intracellular ROS assay), qRT-PCR (quantitative real-time reverse transcription PCR, section Total RNA extraction and qRT-PCR), TRIzol (guanidine thiocyanate, section Total RNA extraction and qRT-PCR), CT (cycle of quantification/qualification of a qPCR), section Total RNA extraction and qRT-PCR, GAPDH (glyceraldehyde 3-phosphate dehydrogenase, section Total RNA extraction and qRT-PCR), Q (quality score, section Transcriptome analysis), GC content (Guanine-cytosine content, section Transcriptome analysis), UHPLC (ultra high performance liquid chromatography, section Metabolomics analysis), PC (phosphytidylcholine, section Changes in cell metabolism after ApoE KO), BEH (section Metabolomics analysis), VIP (section metabolomics analysis), PE (phosphatidyl ethanolamine, section Changes in cell metabolism after ApoE KO), TCA (tricarboxylic acid cycle, section L-Glutamic acid is the central substance of metabolic changes in KO cells), Aldh4a1, (Aldehyde dehydrogenase family 4, member A1, section L-Glutamic acid is the central substance of metabolic changes in KO), Got (glutamic oxaloacetic transaminase, section L-Glutamic acid is the central substance of metabolic changes in KO cells), Gpt2 (Glutamine-pyruvate transaminase 2, section L-Glutamic acid is the central substance of metabolic changes in KO cells), KEGG (Kyoto Encyclopedia of Genes and Genomes, section Decreased expression of Slc7a8 and Slc6a19 is responsible for abnormal L-glutamic acid metabolism in KO cells), GSEA (gene set enrichment analysis, section Decreased expression of Slc7a8 and Slc6a19 is responsible for abnormal L-glutamic acid metabolism in KO cells), GABA (gamma aminobutyric acid, discussion), mTor (mammalian target of rapamycin, member of the phosphatidylinositol 3-kinase-related kinase family of protein kinases, discussion).
f) Include all used abbreviations of the tables and figures into the legends, although already mentioned in the manuscript (Figure 4: PI3K, phosphoinositide 3-kinases, cGMP, cyclic guanosine monophosphate, PKG, protein kinase G, GSH, glutathione, ADP, adenosine diphosphate, UMP, uridine monophosphate, UDP, uridine diphosphate, and mention the meanings of col, collagen, mep, 2-C-Methyl-D-erythritol 4-phosphate, mme, membrane metallo-endopeptidase, atp, adenosine triphosphate, Figure 5: DAPI, 4',6-Diaminidin-2-phenylindol, Table 1: GAPDH, glyceraldehyde 3-phosphate dehydrogenase, PAM, Table 2: F, forward, R, reverse, and mention the abbreviations of GAPDH, Scl7a8, Scl16a9, Figure S4: NAD). Include a description of the genes into Figure S5.
g) Check the references for accuracy according to the Journal Style Guidelines. Check the manuscript for small and capital letter use in the article title.
h) Language: 1. Abstract, line 7, please change to: 'metabolomic'.
2. Section, Decreased expression of Slc7a8 and Slc6a19 is responsible for abnormal L-glutamic acid metabolism in KO cells, change to: 'the protein digestion and absorption pathway'.
3. Section, Metabolomic analysis, line 10 and line 11, check for spacing.
4. Section, Transcriptome analysis, line 10, change to: 'p<='.
5. Results, line 8, check for spacing.
6. Results, line 17, change to: 'the PC1 content'.
7. Discussion, line 1: What is meant by 'the short duration of of atherosclerosis and hyperlipidemia'?
Sincerely,
Author Response
Responding to Reviewer 2
Thank you for your constructive comments on our manuscript. Your comments are valuable for improving our present experiment. The manuscript has been revised accord to your comments. The details of revisions are listed as below.
---------------------------------------------------------------------------------------------------------------
a) Please include a conclusion, wherein parts of the well-discussed results from the discussion section and the last nine lines of the introduction can be shifted.
Answer: The section of Discussion was revised and reorganized as you suggested.
b) Make sure that the letter size in the figures and supplementary figures are high or big enough that it can be read without difficulties in any software format.
Answer: We replaced Fig 1, Fig 3, and Fig 4 with the high-solution images.
c) Explain in the legends the p-values in Figure 3 and the curve of the enrichment score of Figure S6. Is it correct that in Figure 1A component 1 is on both axes and in Figure 1C PC1 and PC2?
Answer: We have added supplementary content to the figure notes in Figures 3 and S6. As you suggested, the two axes in Figure 1A and 1C were corrected.
d) Mention of every software or materials used the producer with its main residence: HEI-OC1 cells (House Ear Instituts-Organ of Corti cells, Los Angeles, USA), Omicsstudio (introduction, line 12 from the bottom), Amaxa basic nucleofector kit (Lonza, Cologne, Germany, section, Generation of ApoE-deficient HEI-OC1 cell lines), KeyGen Biotech (Nanjing, China, section Intracellular ROS assay), Vazyme (Nanjing, China, section, Intracellular ROS assay), Qubit fluorometer (section Transcriptome analysis), Illumina (San Diego, USA, section Transcriptome analysis), DESeq2 R package (section Transcriptome analysis), UHPLC system (Vanquish, Thermo Fisher Scientific, Waltham Massachusetts, USA, section Metabolomics analysis), Orbitrap (section Metabolomics analysis), HMDB (human metabolome database, University of Alberta, Canada, section Changes in cell metabolism after ApoE KO), Metamapp platform (University of California, Davis, USA, section, L-Glutamic acid is the central substance of metabolic changes in KO cells), Cytoscape (Cytoscape Consortium, University of California, San Francisco, USA, University of California, San Diego, USA, University of Toronto, Canada, Gladstone Institutes, San Francisco, USA, Agilent Technologies, Santa Clara, USA, Institut Pasteur, Paris, France, Institute for Systems Biology, Seattle USA, section, L-Glutamic acid is the central substance of metabolic changes in KO cells).
Answer: We have added the manufacturer and address of the software or material used.
e) The manuscript needs to be thoroughly revised for mentioning all abbreviations, when first used: HEI-OC1 (abstract), KO (knockout, abstract), OHC, IHC (outer hair cells, inner hair cells, line 13 of the introduction), ROS (reactive oxygen species, line 3 from the bottom of the introduction), genes Slc7a8, Slc6a19 (explain them already in the introduction and not in the discussion, Slc7a8=solute carrier membrane transport protein, family 6, subfamily a, member 8, large neutral amino acids transporter, Slc6a19=solute carrier membrane protein, family 6, subfamily a, member 19, sodium-dependent neutral amino acid transporter), LC-MS (liquid-tomography-mass-spectrometry, introduction, line 9 from the bottom), FBS (fetal bovine serum, section cell culture), PBS (phosphate buffered saline, section cell culture), Cas9 (CRISPR, clustered regularly interspaced short palindromic repeats, associated protein 9, section Generation of ApoE-deficient HEI-OC1 cell lines), G418 (geneticin, aminoglycoside antibiotic, section Generation of ApoE-deficient HEI-OC1 cell lines), WB (western blot, section Generation of ApoE-deficient HEI-OC1 cell lines), PCR (polymerase chain reaction, section Generation of ApoE-deficient HEI-OC1 cell lines), DCFH-DA (dichlorfluorescein-discrimination analysis), Rosup (ROS stimulating agent, section intracellular ROS assay), EDTA (Ethylenediaminetetraacetic acid, section apoptosis detection), FITC (Fluorescein isothiocyanate, section intracellular ROS assay), PI (section, intracellular ROS assay), qRT-PCR (quantitative real-time reverse transcription PCR, section Total RNA extraction and qRT-PCR), TRIzol (guanidine thiocyanate, section Total RNA extraction and qRT-PCR), CT (cycle of quantification/qualification of a qPCR), section Total RNA extraction and qRT-PCR, GAPDH (glyceraldehyde 3-phosphate dehydrogenase, section Total RNA extraction and qRT-PCR), Q (quality score, section Transcriptome analysis), GC content (Guanine-cytosine content, section Transcriptome analysis), UHPLC (ultra high performance liquid chromatography, section Metabolomics analysis), PC (phosphytidylcholine, section Changes in cell metabolism after ApoE KO), BEH (section Metabolomics analysis), VIP (section metabolomics analysis), PE (phosphatidyl ethanolamine, section Changes in cell metabolism after ApoE KO), TCA (tricarboxylic acid cycle, section L-Glutamic acid is the central substance of metabolic changes in KO cells), Aldh4a1, (Aldehyde dehydrogenase family 4, member A1, section L-Glutamic acid is the central substance of metabolic changes in KO), Got (glutamic oxaloacetic transaminase, section L-Glutamic acid is the central substance of metabolic changes in KO cells), Gpt2 (Glutamine-pyruvate transaminase 2, section L-Glutamic acid is the central substance of metabolic changes in KO cells), KEGG (Kyoto Encyclopedia of Genes and Genomes, section Decreased expression of Slc7a8 and Slc6a19 is responsible for abnormal L-glutamic acid metabolism in KO cells), GSEA (gene set enrichment analysis, section Decreased expression of Slc7a8 and Slc6a19 is responsible for abnormal L-glutamic acid metabolism in KO cells), GABA (gamma aminobutyric acid, discussion), mTor (mammalian target of rapamycin, member of the phosphatidylinositol 3-kinase-related kinase family of protein kinases, discussion).
Answer: We have amended it accordingly by adding the full name of the abbreviation when it is first used in the manuscript.
f) Include all used abbreviations of the tables and figures into the legends, although already mentioned in the manuscript (Figure 4: PI3K, phosphoinositide 3-kinases, cGMP, cyclic guanosine monophosphate, PKG, protein kinase G, GSH, glutathione, ADP, adenosine diphosphate, UMP, uridine monophosphate, UDP, uridine diphosphate, and mention the meanings of col, collagen, mep, 2-C-Methyl-D-erythritol 4-phosphate, mme, membrane metallo-endopeptidase, atp, adenosine triphosphate, Figure 5: DAPI, 4',6-Diaminidin-2-phenylindol, Table 1: GAPDH, glyceraldehyde 3-phosphate dehydrogenase, PAM, Table 2: F, forward, R, reverse, and mention the abbreviations of GAPDH, Scl7a8, Scl16a9, Figure S4: NAD). Include a description of the genes into Figure
Answer: We have added the full names of all abbreviations in the figure legends.
g) Check the references for accuracy according to the Journal Style Guidelines. Check the manuscript for small and capital letter use in the article title.
Answer: We checked the accuracy of the references and the title of the article.
h) Language: 1. Abstract, line 7, please change to 'metabolomic'.
Answer: Thank you for your advice. We have corrected the mistake.
2. Section, Decreased expression of Slc7a8 and Slc6a19 is responsible for abnormal L-glutamic acid metabolism in KO cells, change to: 'the protein digestion and absorption pathway'.
Answer: According to your suggestion, we have changed the inappropriate title.
3. Section, Metabolomic analysis, line 10 and line 11, check for spacing.
Answer: Thank you for your advice. We have corrected the mistake.
4. Section, Transcriptome analysis, line 10, change to: 'p<='.
Answer: We have modified the description of the p-value in this paragraph.
5. Results, line 8, check for spacing.
Answer: Thank you for your advice. We have corrected the mistake.
6. Results, line 17, change to: 'the PC1 content'.
Answer: Thank you for your advice. We have corrected the mistake.
7. Discussion, line 1: What is meant by 'the short duration of atherosclerosis and hyperlipidemia'?
Answer: This impropriate description was revised in Paragraph 1 of Discussion section.
Round 2
Reviewer 2 Report
Dear Ladies and Gentlemen, Dear Journal-Team,
the authors further increased the quality of their interesting manuscript 'Metabolic abnormalities linked to auditory pathways in ApoE-knockout HEI-OC1 cells: a transcription-metabolism co-analysis. The remaining corrections are listed below.
a) The manuscript still needs a conclusion to improve the understanding of the topic.
b) Please explain the abbreviation DESeq2R package (differential expression analysis for sequence data version 2R, which software suite?, section Transcriptome analysis), BEH and VIP in the section Metabolomics analysis. The terms in Figure S5: Riken Research Institue, Japan, DOD, D-aspartare oxidase.
b) Please, describe the P-value in Figure 3A. Further describe the course of the enrichment score curve in S6. The well defined ups and downs are still difficult to understand.
c) Language: 1. Abstract and section Cell culture, please change to: 'House Ear Institute-Organ of Corti Cells'.
2. Section Intracellular ROS assay, line 97, change to: 'a ROS detection test'.
3. Section Metabolomics analysis, line 151, check for spacing before the parentheses use.
4. Section 3.4, stay with the old headline 'Decreased expression of Slc7a8 and Slc6a19 is responsible for abnormal L-glutamat acid metabolism in KO cells', change in line 285 to: 'the protein digestion and absorption pathway'.
5. Section Decreased expression of Slc7a8 and Slc6a19 is responsible for abnormal L-glutamat acid metabolism in KO cells, line 293, change to: 'by qPCR'.
6. Discussion, line 374, change to: 'while an altered glutamate content'.
7. Figure 4 legend, change to: 'phosphorinositide 3-kinases', and check the further figure legend for accuracy.
8. Figure S5 legend, change to: 'glutamine-fructose-6-phosphate transaminase 2'.
Sincerely,
Author Response
Responding to Reviewer 2
Thank you for your constructive comments on our manuscript. Your comments are valuable for improving our present experiment. The manuscript has been revised accord to your comments. The details of revisions are listed as below.
---------------------------------------------------------------------------------------------------------------
- a) The manuscript still needs a conclusion to improve the understanding of the topic.
Answer: As your suggestion, a conclusion has been added to the last paragraph of the Discussion
- b) Please explain the abbreviation DESeq2R package (differential expression analysis for sequence data version 2R, which software suite?, section Transcriptome analysis), BEH and VIP in the section Metabolomics analysis. The terms in Figure S5: Riken Research Institue, Japan, DOD, D-aspartare oxidase.
Answer: According to your suggestion, we have added the full name of the abbreviation and the term of the picture.
- c) Please, describe the P-value in Figure 3A. Further describe the course of the enrichment score curve in S6. The well defined ups and downs are still difficult to understand.
Answer: According to your suggestion, we have further described the corresponding content.
- d) Language: 1. Abstract and section Cell culture, please change to: 'House Ear Institute-Organ of Corti Cells'.
Answer: Thank you for your advice. We have corrected the mistake.
- Section Intracellular ROS assay, line 97, change to: 'a ROS detection test'.
Answer: Thank you for your advice. We have corrected the mistake.
- Section Metabolomics analysis, line 151, check for spacing before the parentheses use.
Answer: Thank you for your advice. We have corrected the mistake.
- Section 3.4, stay with the old headline 'Decreased expression of Slc7a8 and Slc6a19 is responsible for abnormal L-glutamat acid metabolism in KO cells', change in line 285 to: 'the protein digestion and absorption pathway'.
Answer: Thank you for your advice. We have corrected the mistake.
- Section Decreased expression of Slc7a8 and Slc6a19 is responsible for abnormal L-glutamat acid metabolism in KO cells, line 293, change to: 'by qPCR'.
Answer: Thank you for your advice. We have corrected the mistake.
- Discussion, line 374, change to: 'while an altered glutamate content'.
Answer: Thank you for your advice. We have corrected the mistake.
- Figure 4 legend, change to: 'phosphorinositide 3-kinases', and check the further figure legend for accuracy.
Answer: Thank you for your advice. We have corrected the mistake.
- Figure S5 legend, change to: 'glutamine-fructose-6-phosphate transaminase 2'.
Answer: Thank you for your advice. We have corrected the mistake.